# Three-Dimensional Structural Stability and Local Electrostatic Potential at Point Mutations in Spike Protein of SARS-CoV-2 Coronavirus

**DOI:** 10.3390/ijms25042174

**Published:** 2024-02-11

**Authors:** Svetlana H. Hristova, Alexandar M. Zhivkov

**Affiliations:** 1Department of Medical Physics and Biophysics, Medical Faculty, Medical University—Sofia, Zdrave Street 2, 1431 Sofia, Bulgaria; svhristova@medfac.mu-sofia.bg; 2Scientific Research Center, “St. Kliment Ohridski” Sofia University, 8 Dragan Tsankov Blvd., 1164 Sofia, Bulgaria

**Keywords:** SARS-CoV-2, coronavirus variants, point mutations, S-protein, ACE2 receptor, protein electrostatics, isoelectric point, surface electric potential, folding energy, contagiousness

## Abstract

The contagiousness of SARS-CoV-2 β-coronavirus is determined by the virus–receptor electrostatic association of its positively charged spike (S) protein with the negatively charged angiotensin converting enzyme-2 (ACE2 receptor) of the epithelial cells. If some mutations occur, the electrostatic potential on the surface of the receptor-binding domain (RBD) could be altered, and the S-ACE2 association could become stronger or weaker. The aim of the current research is to investigate whether point mutations can noticeably alter the electrostatic potential on the RBD and the 3D stability of the S1-subunit of the S-protein. For this purpose, 15 mutants with different hydrophilicity and electric charge (positive, negative, or uncharged) of the substituted and substituting amino acid residues, located on the RBD at the S1-ACE2 interface, are selected, and the 3D structure of the S1-subunit is reconstructed on the base of the crystallographic structure of the S-protein of the wild-type strain and the amino acid sequence of the unfolded polypeptide chain of the mutants. Then, the Gibbs free energy of folding, isoelectric point, and pH-dependent surface electrostatic potential of the S1-subunit are computed using programs for protein electrostatics. The results show alterations in the local electrostatic potential in the vicinity of the mutant amino acid residue, which can influence the S-ACE2 association. This approach allows prediction of the relative infectivity, transmissibility, and contagiousness (at equal social immune status) of new SARS-CoV-2 mutants by reconstruction of the 3D structure of the S1-subunit and calculation of the surface electrostatic potential.

## 1. Introduction

SARS-CoV-2 (originator of COVID-19 pandemic) is a β-coronavirus which infects the human epithelial cells of the respiratory, cardiovascular, and excretory systems. The virus particles penetrate into the cells after the association of its spike (S) protein with some integral membrane proteins [1]; the most investigated is the angiotensin-converting enzyme (ACE2) whose main function is the reduction in the blood pressure by the detachment of one amino acid residue from the inactive decapeptide angiotensin-1 and its derivate angiotensin-2 (octapeptide, the strongest vascular constrictor); this membrane enzyme is canonically pointed out as a receptor for the S-protein. After S-ACE2 noncovalent binding, the coronavirus particles are absorbed by receptor-mediated endocytosis [2]. About 20 viruses of the *Coronaviridae* family cause infections predominantly of the gastro-intestinal tract of animals (diarrhea of cows, etc.) and light human respiratory infections. The appearance of a stronger-infecting strain of SARS-CoV-2 is caused by point mutations in the viral RNA and corresponding changes in its S-protein.

The S-protein is a large homotrimer constructed by three polypeptide chains, with each of them forming two subunits (artificially divided by the enzyme plasmin which cuts the chains at residue N685) [3]: S1 (hydrophilic extramembrane, consisting of 672 amino acid residues) and S2 (588 residues which have massive hydrophilic extramembrane and small hydrophobic intramembrane parts). The S-protein is integrated into the viral lipid membrane (which envelopes the RNA–protein complex) by the three S2-subunits, but associates with the ACE2 receptor using one of three large water-soluble S1-subunits. The alterations in the coronavirus infectivity are caused by point mutations in the receptor-binding domain (RBD, consisting of 195 amino acid residues, with 31 of them located on its surface), which is part of the S1-subunit [4]. The comparison of the amino acid sequences of the S1-subunits of SARS-CoV-2 and its less infecting predecessor SARS-CoV-1 (isolated in 2003) reveals that the drastically increased contagiousness of the newer strain is determined by change in RBD at which, instead of a vanished uncharged valine amino acid residue (V404), one positively charged lysine (417 K) emerges [5], which meets an oppositely (negatively) charged aspartate amino acid residue of the ACE2 receptor [6], leading to stronger protein–protein electrostatic attraction; as a result of this charge-chaining point mutation, the constant of association of S1 to ACE2, measured by surface plasmon resonance, is 7.5 times higher in the case of SARS-CoV-2 in comparison with SARS-CoV-1 [4]. On the other hand, this assertion is supported by the supposition that other factors (natural resistance, adaptive, and artificial individual and group immunity, etc.), which also determine the infectivity, transmissibility, and contagiousness, were approximately equal for SARS-CoV-1 and SARS-CoV-2 when the last mutant was isolated in the south-east region of China.

The influence of the electrostatic forces on the affinity of the S1-subunit to the ACE2 receptor is also confirmed experimentally in the cases of mutations K417N and K417T, at which the positively charged amino acid residue lysine (K) is replaced with uncharged asparagine (N) or threonine (T), and at mutation E484K, where a negatively charged glutamic acid (E) is replaced with positively charged lysine. As a result of these charge-changing point mutations of SARS-CoV-2, the S1-ACE2 constant of association, measured by surface plasmon resonance, decreases or increases, respectively [7]. These experimental results confirm the supposition that the electric forces between the positively charged S1-subunit and the negatively charged ACE2 receptor are important factors for their association [8,9,10].

The effects of point mutations on the electric properties of the S-protein and the S-ACE2 association have been investigated by many authors employing different methods and using different protein structures [11,12,13,14,15,16,17,18,19,20,21,22,23,24,25,26,27,28,29,30,31,32]. In most cases, the studies consider the RBD or its fragment in monomeric form, the RBD-ACE2 association is explained by pairs of positive–negative electric charges oppositely located on the two proteins, and the ionization of the chargeable groups is assumed to be constant, neglecting their dependence on pH and the 3D structure of the protein globule. In our opinion, the most correct electrostatic simulations have been carried out by Barosso da Silva et al. [32], where the pH-dependent net electric charge of the RBD of 19 coronavirus mutants is calculated considering the 3D-dependent dissociation constants pK_a_ of the ionizable groups (which determine the net charge at a given pH); in this investigation, a linear correlation of the RBD-ACE2 binding affinity on the net electric charge of the RBD of the wild-type and five coronavirus variants is found (only Omicron variant shows some deviation from the linear dependence), which allows prediction of the infectivity of new coronavirus mutants.

The direct relation between the 3D molecular structure of the S1-subunit and its constant of association to the human ACE2 receptor makes it possible to estimate the potential of charge-changing point mutations in the S-protein to increase or decrease the relative infectivity, contagiousness, and viral spread of coronavirus variants and their subvariants. This can help predict the possibility of new coronavirus epidemiological waves under similar conditions. Such predictions can be made once the amino acid sequence of the unfolded polypeptide chain of the mutants has been determined using biochemical methods. To do this, as a first step, it is necessary to reconstruct the 3D structure of the mutant S1-subunit or the whole S-protein, and then calculate the electric charges on the surface of the RBD at the RBD-ACE2 binding interface. Since molecular dynamic simulations are limited by the polymer chain length, authors are forced to use the RBD (a polypeptide consisting of only two hundred amino acid residues) instead of the S1-subunit when performing 3D reconstruction. We employ a computer technique, which allows the use of the longer polypeptide chain of the S1-subunit. This technique is based on determining the atom coordinates at which the protein globule has minimal free energy. The results obtained in our previous investigation [33] have shown that the pH-dependent S1-ACE2 binding energy is different for the wild-type coronavirus and Omicron variant. This reveals the secret why this mutant is more infective but less pathogenic: at pH 5–6, the binding of the Omicron S-protein to the ACE2 receptors of the epithelial cells in the upper respiratory tract is stronger compared to the binding in the blood vessels, where the pH of the blood plasma is 7.4.

Our approach is based on the understanding that not the distinct charges of the amino acid residues but the local electric potential, created by all charges of the protein globule, is the main driving force for the protein–protein association. Therefore, we calculate the local potential at the RBD-ACE2 binding interface and its alteration upon point mutations emerging in the RBD region, considering the 3D coordinates of all pH-dependent charges of the S1-subunit. The importance of the electrostatic forces for virus–receptor binding follows from the fact that the replacement of one amino acid residue by another does not usually noticeably change the 3D structure of the S-protein, as confirmed by the present computer investigation, and consequently, the relief of the contacting protein–protein surfaces remains unchanged. As a result, the other forces (van der Waals, hydrogen bonds, hydrophobicity), which contribute to the S-ACE2 association, remain almost unaltered; this is conditioned by the small size of the mutant amino acid residue in comparison with the relatively large area at the contact interface of the RBD.

To estimate the influence of a single charge-changing point mutation on the surface electrostatic potential at the binding interface of the S-protein in the present computer investigation, we analyze 15 SARS-CoV-2 point mutants which have been selected from a large number of mutants using the criterion that the initial and substituting amino acid residues are situated on the surface of the S1-subunit in the region of the RBD and have different charge (positive, negative, or uncharged) or different hydrophilicity/hydrophobicity. The aim is to estimate alterations in (a) the stability of the 3D structure of the S1-subunit of the mutants, and (b) the electrostatic potential on the surface at the S-ACE2 receptor-binding interface, which determines the intermolecular protein–protein electric attractive/repulsive forces created by all pH-dependent electric charges of the two protein globules.

As the first step, we reconstruct the 3D structure of the S1-subunit of a chosen coronavirus variant, starting from the published crystallographic structure of the S-protein of the wild-type coronavirus and the amino acid sequence of the polypeptide chain of the mutant S-protein. Through this process, we obtain the coordinates of all its atoms, which are needed to calculate the electric potential created by the coulomb electric charges of the protein globule. After the reconstruction of the 3D structure of the mutants, employing the methods of protein electrostatics, we calculate the isoelectric point, the surface electrostatic potential, and the folding free energy of the S-subunit at a given pH of the medium. The results indicate that the change in even a single coulomb charge at the binding interface of the RBD causes a significant alteration of its local surface electrostatic potential, but other parameters do not undergo noticeable alterations. The main inference of our work is that the local electrical potential on the RBD of the S-protein is the actual criterion for predicting whether a point mutation will increase or decrease the infectivity of a new coronavirus variant, rather than its isoelectric point, as researchers in the field believe. The developed approach can be applied to other viral or non-viral proteins that undergo point mutations.

## 2. Results and Interpretation

### 2.1. ACE2 Receptor

The calculations give a pI of 5.11 for the isoelectric point and Δ*G*_fold_ = −292.69 J/mol for the free energy of folding at pH 6–7 of the extramembrane domain of the ACE2 receptor of the human epithelial cells. The surface electrostatic potential at the virus-binding interface is negative. This local potential is conditioned by 79 negative and 52 positive charges of the amino acid residues, whose chargeable groups have either a dissociated or associated proton (H^+^-ion). The negative sign of the surface potential electrostatically facilitates the association of the ACE2 receptor with the positively charged S1-subunit of the S-protein (Figure 1).

### 2.2. Point Mutants of S-Protein

Fifteen variants of SARS-CoV-2 β-coronavirus have been selected to evaluate diverse types of alterations (compared to the wild-type strain). These changes are caused by the substitution of one amino acid residue with another at point mutations in the polypeptide chain of the S-protein (Figure 2). The selection criteria include the following: (a) the substituted and substituting (mutant) residues are located on the contact surface of the RBD by which the S1-subunit associates with the ACE2 receptor of the epithelial cells (Figure 1); (b) the residues have direct contact with the water environment, as they are located on the surface of the protein globule or in a crypt pocket. The chosen mutants (Table 1) represent three types of electric and thermodynamic changes: (a) substitution with a change in the electric charges (at least one of the replaced or substituting amino acid residue is charged at pH 5–7, both are hydrophilic); (b) substitution with a change in both charge and hydrophilicity; and (c) substitution with a change in the hydrophilicity/hydrophobicity of uncharged amino acid residues. The hydrophilicity is determined by the alteration of the Gibbs free energy Δ*G*_trans_ when an amino acid residue is transferred between media with different dielectric permittivity ε (a measure of hydrogen bonds which leads to the formation of molecular aggregates with a high permanent dipole moment): from ethyl alcohol (ε = 25 at 20 °C) to water medium (ε = 80). A negative Δ*G*_trans_ (Table 1) means that the given residue is hydrophilic, and a positive Δ*G*_trans_ means that it is hydrophobic. The values of Δ*G*_trans_ are taken from Ref. [34].

The reconstruction of the mutant models has been accomplished using the 3D atomic coordinates of the S1-subunit of the S-protein of the wild-type strain of the β-coronavirus SARS-CoV-2 (crystallographic structure PDB: 6LZG). Although the atomic coordinates are slightly shifted in the region of the point mutations, the aliment of the polypeptide chains of the wild-type strain and the coronavirus variants indicates that the chosen point mutations do not noticeably alter the 3D structure of the S1-subunit. The two chains visibly coincide, with only minor changes in the region of the mutations (the upper rows in Figure 3, Figure 4, Figure 5, Figure 6 and Figure 7). A significant local alteration emerges only in the case of a replacement of the amino acid proline owing to its specific structure. The reason for the absence of significant distortion to the polypeptide backbone is that the replaced and substituting amino acid residues have relatively small molecular volumes (Table 1). This allows formation of the same secondary structure of the polypeptide chain (α-helix and β-sheet).

### 2.3. Isoelectric Point

The electric charge of the protein globule is determined by the pH-dependent ionization of the amino and carboxylic groups of the amino acid residues located on the surface of the globule or in the pocket where they make contact with the water environment. In the second rows of Figure 3, Figure 4, Figure 5, Figure 6 and Figure 7, the substituted (the first pictures) and the substituting (the second, third, and fourth pictures) amino acid residues are colored according to their charge and hydrophilicity: green (uncharged hydrophilic); red (negatively charged hydrophilic); blue (positively charged hydrophilic); and yellow (uncharged hydrophobic).

The calculated isoelectric point pI (pH at which there is a time-averaged equality of positive and negative coulomb charges of the protein macromolecule) of the S1-subunit and its RBD of 15 different mutations in the spike protein are all shown in Table 2. The shift ΔpI in the isoelectric point is defined as the difference between the pI values of the mutant and the wild-type strain:ΔpI = [pI]_mutant_ − [pI]_wild_
(1)

The values of ΔpI of the mutants (Table 2) indicate that even a single charge-changing point mutation in the polypeptide chain can significantly shift the isoelectric point of the S1-subunit in its 3D native conformation, although one proton is only 5.9% of the 17 positive coulomb charges of the amino groups (–NH_3_^+^), or accordingly, one electron compared to the 17 negative charges of the carboxylic groups (–COO^–^) at the isoelectric point. Considering that the isoelectric point of the S1-subunit is close to pI 9, it can be concluded that the pI-shift is caused by alterations in the degree of ionization of the amino groups (–NH_3_^+^ ↔ –NH_2_) because, in this pH range, all carboxylic groups are fully ionized.

However, the isoelectric point pI of the S1-subunit or its RBD could not be used as a reliable criterion for prediction of the infectivity of the new coronavirus variants, because pI (Table 2) reflects the zero net charge of all chargeable groups at pH 9, but in the physiologically important range pH 6–7, the summed charges are far from equal. The pI value of 9 indicates that the net charge is positive at neutral pH, but its actual value remains unknown because the net charge is determined by the numbers of charged carboxylic and amino groups. Therefore, the pI value could not be used for quantitative prediction of the total charge. The force acting when two protein globules approach each other is the electrostatic potential created by all their coulomb charges, but the contribution of the individual charges depends on the distance between the globules. At a long distance, the electrostatic attraction of the S-protein to the negatively charged ACE2 receptor is determined by their net charge. At short distance, the leading factor is the electrostatic potential created by the charges located at the contacting interface. Therefore, to predict the influence of point mutations on the S-ACE2 association constant, and thus the alterations of their transmissibility, it is necessary to calculate the local electrostatic potential at the S-ACE2 binding interface at a given pH, considering the irregular distribution of the charges on the surface of the protein globules.

### 2.4. Local Electrostatic Potential

The electrostatic potential φ on the surface of the receptor-binding interface of the wild-type strain (the left pictures in the third rows of Figure 3, Figure 4, Figure 5, Figure 6 and Figure 7) and the mutants (the next three pictures in the same rows) is visualized based on its sign and intensity. The pictures display small parts of the surface of the RBD of the S-protein in the region of the point mutations; the replaced and substituting amino acid residues are denoted by cycles. The surface of the models is colored according to the electric potential φ = *kT*/*e*, in blue (positive), red (negative), and white (neutral) with the scale ± 6 *kT*/*e* (1 *kT*/*e* [J/C] = 26.7 mV at 37 °C). The pictures show that the point mutations emerge in regions with different electrostatic potential: strongly positive (Figure 3), slightly negative (Figure 4), slightly positive (Figure 5), and almost neutral (Figure 6). The different potential in the vicinity of the substituted amino acid residue and the sign of the substituting one lead to different results: from the alteration in the value of the potential only (Figure 3, Figure 6 and Figure 7) to the change in its sign (Figure 4 and Figure 5). These alterations of the local electric potential influence the association constant of binding of the S-protein to the ACE2 receptor because only the potential at the S-ACE2 binding interface determines the protein–protein electrostatic interactions, rather than its averaged value over the entire surface (the latter is zero at the isoelectric point pI).

### 2.5. Free Energy of Folding

The Gibbs free energy of folding Δ*G*_fold_ is defined as a difference between two states of the polypeptide chain: folded (native 3D protein structure) and unfolded (random coil conformation at full denaturation). We define the difference between the Δ*G*_fold_ values of the mutant and the wild-type strain as the free energy increment ΔΔ*G*_fold_, which can be either positive or negative:ΔΔ*G*_fold_ = [Δ*G*_fold_]_mutant_ − [Δ*G*_fold_]_wild_(2)

#### 2.5.1. Folding Free Energy of S1-Point Mutants

The values of Δ*G*_fold_ and ΔΔ*G*_fold_ (Table 3) are calculated at three pH values of the medium; the first two pH values have physiological significance. pH 6.0 corresponds to the pH of the extracellular secretions in the respiratory tract, where the virus particles diffuse before association with the ACE2 receptors of the epithelial cells. The pH of the nasal and pharynx mucosa is within the range of 5.5–6.5 at the norm [36]. pH 7.0 is close to pH 7.4 of the blood plasma and the cytoplasm. This pH determines the intracellular stability of the S-protein in the process of synthesis and folding of its polypeptide chain in infected epithelial cells and upon following the circulation of the virus particles in blood vessels where they can associate with the ACE2 receptors on vascular epithelial cells. At pH 6–7, the S1-subunit is positively charged, and in this case, the alteration of a single charge has a significantly smaller impact than at the isoelectric point, which is close to pH 9 (Table 2).

The data in Table 3 indicate that the folding energy Δ*G*_fold_ of the mutants (except N2) is somewhat decreased in absolute value (with the maximum difference being 15% at pH 6) compared to that of the wild-type strain. However, Δ*G*_fold_ remains negative, indicating that the 3D structure of the S1-subunit remains stable despite these point mutations. The almost equal values of the increment ΔΔ*G*_fold_ at pH 7.0 and pH 6.0 reflect the fact that the degree of ionization of the chargeable groups does not change even with a tenfold alteration in H^+^-ion concentration in this pH range [37].

Three factors can contribute to the alteration ΔΔ*G*_fold_ of the folding energy Δ*G*_fold_ caused by a point mutation: electrostatic attraction/repulsion between the charges of the polypeptide chain, hydrophilicity/hydrophobicity, and the molecular volume of the replaced/substituting amino acid residues.

#### 2.5.2. Isoelectric Point as an Indicator for 3D Structural Stability

The comparison of the data in Table 2 and Table 3 suggests that the change in the coulomb electric charges of the S1-subunit (Figure 2) is one of the factors responsible for the alteration ΔΔ*G*_fold_ of the folding free energy Δ*G*_fold_ at point mutations when the substituted and/or substituting amino acid residues are located on the surface of the protein globule dissolved in an aqueous medium. The increment ΔΔ*G*_fold_ is pH-dependent because the coulomb electric charges, located at the protein/water interface, are determined by protonation/deprotonation of the amino and carboxylic groups of the chargeable amino acid residues. The shift in the net charge from the electrically neutral state (pI), independently of whether the additional electric charge is positive or negative (the sign of ΔpI in Table 2), should be accompanied with decreased 3D stability (smaller absolute value |Δ*G*_fold_|, positive ΔΔ*G*_fold_) since the folding of the polypeptide chain into a globule is somewhat hindered owing to the electrostatic repulsion between the identically charged amino acid residues. At pH 6–7, the net charge of the S1-subunit of the wild-type strain is positive (the isoelectric point is pI 8.7, Table 2), so the emergence of a negative coulomb charge in its polypeptide chain should increase the 3D stability of the protein globule, and vice versa: the stability should decrease when the additional charge is positive.

The influence of the electric charges can be clearly seen in the case of mutant N354D: the alteration of the folding free energy Δ*G*_fold_ is caused by the substitution of one uncharged amino acid residue (asparagine) with a negatively charged (aspartate) residue which has the same molecular volume, and both residues are hydrophilic with a similar energy of hydration (N2 in Table 1); the substitution causes a shift in the isoelectric point of 0.2 pH units to a lower value (Table 2). In this case, the mutant has a more stable 3D structure of the S1-subunit, as indicated by the increased absolute value |ΔΔ*G*_fold_| (negative increment ΔΔ*G*_fold_, Table 3) and the negative values of the pH dependence ΔΔ*G*_fold_(pH) (Figure 8, curve N354D). The increased structural stability can be explained by reduced electrostatic repulsion in the S1-globule, which is caused by the emergence of one negative coulomb charge in its positively charged (in the range under pH 8.7) polypeptide chain.

In the above case (mutant N354D), the expectation that the decrease in the net charge of the polypeptide chain will stabilize the 3D structure of the protein globule is confirmed by the increase in the absolute value |Δ*G*_fold_| of the folding free energy (negative ΔΔ*G*_fold_). When the same uncharged amino acid residue (asparagine) of the wild S1-subunit is substituted with the positively charged arginine residue in mutant N354R (ΔpI 0.17, N1 in Table 2), the 3D structural stability expectedly decreases (smaller absolute value |Δ*G*_fold_|, positive ΔΔ*G*_fold_ in Table 3 and Figure 8, curve N354R) because of the stronger electrostatic repulsion in the protein globule owing to the increased net (positive) charge. However, upon substitution of the positive arginine with the negative aspartate in the mutant R408D (N7 in Table 2), or with the neutral asparagine (R408N, N8), the 3D structural stability of the S1-subunit also decreases (reduced |Δ*G*_fold_|, positive ΔΔ*G*_fold_, N7 in Table 3), although in these cases, the net charge of the polypeptide chain decreases. A decrease in the 3D stability also appears in the case of substitution of the positively charged arginine with the neutral isoleucine in the mutant R408I (N9 in Table 2 and Table 3).

Therefore, the rule formulated above is not valid: the 3D structural stability of the protein globule (indicated by the absolute value |Δ*G*_fold_| of the folding free energy) increases as the net charge of the polypeptide chain decreases. This means that the isoelectric point pI should not be used as a criterion for prediction of the 3D stability of the S-protein at point mutations. This inference can be extended to other globular proteins because the α-helix and the unstructured segments of the polypeptide chain of the coronavirus S-protein are dimensionally fixed by the same forces: hydrophobic, electrostatic, and hydrogen bonds.

#### 2.5.3. Effect of the Local Electrostatic Potential

The above reasoning regarding the relation between a coulomb charge excess in a polypeptide chain and the 3D structural stability of a protein globule could be valid if the electric charges are evenly distributed along the polyelectrolyte chain, as in synthetic polymers. However, in every globular protein, the charged amino acid residues are irregularly located, depending on both the amino acid sequence in the chain and its 3D conformation. This leads to local electric fields with different (in sign and value) electrostatic potential on the surface of the protein globule, which is a superposition of the elementary electric fields of all coulomb charges in the protein globule (their contribution is determined by the distance; i.e., the neighboring electric charges have stronger influence). It is reasonable to suppose that the alteration (increasing or decreasing) of the folding free energy Δ*G*_fold_ is determined by the local electrostatic potential where the mutant amino acid residue is located. When an extra charge emerges in a region with an opposite electrostatic potential, this should cause an increase in the 3D structural stability because of a decrease in the electrostatic repulsion between the neighboring charges, and vice versa: the stability decreases when the additional charge has the same sign as the local electric potential. To verify this supposition, the cases when a mutant electric charge has the same or opposite sign to the surface electrostatic potential, where it is located, can be considered.

In the case of the mutant N354D (N2 in Table 2), the additional negative charge appears in a positively charged region with a high electrostatic potential (the pictures in the second and third rows of the columns Asn354 and 354Asp in Figure 3). This causes a decrease in the local potential and an increase in the 3D structural stability of the S1-subunit, as indicated by the increased absolute value |ΔΔ*G*_fold_| (negative increment ΔΔ*G*_fold_, Table 3). Because this mutation emerges at the S1-ACE2 binding interface (Figure 2), it leads to a decrease in the constant of the virus–receptor association; i.e., the point mutation N354D causes an increased 3D structural stability of the S-protein, but a lower infectivity of this coronavirus variant. It is interesting that the maximal 3D structural stability appears at pH 5 (the deeper minimum of curve N354D in Figure 8), which corresponds to the pH in the upper respiratory tract.

At mutation N354R (N1 in Table 2), the strong positive local electrostatic potential increases due to the substitution of the neutral asparagine residue with the positively charged arginine (the pictures Asn354 and 354Arg in the third row in Figure 3). As a result, the 3D structural stability of the S1-subunit expectedly decreases (reduced absolute value |ΔΔ*G*_fold_|, positive increment ΔΔ*G*_fold_, Table 3) because of the stronger electrostatic repulsion between the neighboring coulomb charges.

In the case of the substitution of the negatively charged aspartate residue, located in a region with weak negative potential (Asp364 column in Figure 4), with the positively charged arginine (mutant D364R, N4 in Table 2) or with the neutral asparagine (D364N, N5), the local electrostatic potential changes its sign from slightly negative to positive or almost neutral, respectively (the pictures 364Arg and 364Asn in the third row in Figure 4). This leads to a decrease in the 3D structural stability of the S1-subunit (indicated by decreased absolute value |Δ*G*_fold_| of the folding free energy, N4 and 5 in Table 3). So, these point mutations confirm the leading role of the local electrostatic potential in the alteration of 3D structural stability of the S1-subunit.

Upon substitution of the positively charged arginine residue (the Arg408 column in Figure 5) with the negatively charged aspartate residue (mutant R408D, N7 in Table 2), or with the neutral asparagine (mutant R408N, N8), the local electrostatic potential changes its sign from slightly positive to slightly negative or neutral, respectively (the columns 408Asp and 408Asn). This should lead to an increase in the 3D structural stability of the S1-subunit, but the decrease in the absolute value |Δ*G*_fold_| of the folding free energy (positive ΔΔ*G*_fold_, Table 3) disproves this expectation. These last two point mutants reveal that, with small changes in the local electrostatic potential, it is not the leading factor determining the 3D structural stability of the protein globule.

#### 2.5.4. Surface Hydrophilic Effect

In the above cases of point mutations (Section 2.5.3), both the substituted and the substituting amino acid residues are hydrophilic, and only their electric charges are different (N1–9 in Table 1 and Table 2). As a rule, on the surface of a protein globule, the most amino acid residues are hydrophilic, while the hydrophobic residues are located in the core of the protein globule. This rule is also satisfied in the case of the S-protein (except for the short intramembrane hydrophobic segment of the S2-subunit). Because point mutations occur accidentally during intracellular syntheses of viral RNA, it is probable that the mutant amino acid residue will be both hydrophilic and hydrophobic. When the substitution emerges on the surface of the protein globule (in the present research, we consider only mutations located on the surface of the S1-subunit) and a hydrophilic amino acid residue is replaced with a hydrophobic one, the 3D structural stability should decrease.

To estimate the contribution of local hydrophilicity/hydrophobicity as a second factor contributing to the 3D stability of the S1-subunit, the free energy of folding Δ*G*_fold_ must be calculated for point mutants whose substituted/substituting residues are uncharged but have different hydrophilicity (estimated by alteration Δ*G*_trans_ of Gibbs free energy upon transferring from ethanol to water, Table 1). These criteria are satisfied by the mutants N354I (asparagine/isoleucine), W436N (tryptophan/asparagine), and P491N (proline/asparagine) (respectively, N3, N12, and N15 in Table 1 and Table 2).

At the point mutant N354I, the hydrophilic asparagine residue is substituted with the hydrophobic isoleucine (the difference in Δ*G*_trans_ is ΔΔ*G*_trans_ = +13.3 kJ/mol, N3 in Table 1). This is the clearest case because both amino acid residues are uncharged and have equal molecular volume *V*_res_ (the third factor which can contribute to the 3D structural stability of the protein globule). The decreased absolute value |Δ*G*_fold_| of the folding free energy (N3 in Table 3) and the positive ΔΔ*G*_fold_ values of the pH-curve N354I (Figure 8) confirm the expectation that the 3D stability of the S1-subunit should decrease as a result of the substitution of a hydrophilic with a hydrophobic amino acid residue (when they are located on the surface of the protein globule). The higher ΔΔ*G*_fold_ values of the curve N354I in comparison with N354R (Figure 8) indicate that the appearance of the hydrophobic uncharged isoleucine 354I residue on the hydrophilic surface of the S1-subunit (the pictures 354Ile in Figure 3) decreases the 3D structural stability of the protein globule to a greater degree than the substitution of the positively charged hydrophilic arginine residue 354R located in a region with a positive local potential (354Arg, Figure 3).

The 3D stability of the S1-subunit should increase when the hydrophobic tryptophan residue W436 is substituted by the hydrophilic asparagine 436N, considering that the isoelectric point is not shifted (ΔpI 0), because both residues are uncharged (mutant N12 in Table 2). However, the 3D structural stability unexpectedly decreases (positive ΔΔ*G*_fold_, Table 3). This can be explained by a change in the local 3D structure of the S1-subunit (the picture 436Asn, the first row in Figure 6) because of two different molecular volumes *V*_res_ of the two amino acid residues (Table 1).

#### 2.5.5. Deep Electrostatic–Hydrophilic Effect

The highest decrease in the 3D structural stability of the S1-subunit appears at mutant P491R: the alteration of the Gibbs free energy Δ*G*_fold_ is ΔΔ*G*_fold_ ≈ +13 kJ/mol (mutant N13 in Table 3). This is a result of the combined effect of two factors: (a) increased electrostatic repulsion caused by the substitution of the uncharged proline residue P491 with the positively charged arginine residue 491R, which increases the positive local potential (Figure 7, mutant 491Arg), and (b) reverse hydrophilic effect caused by the substitution of the hydrophobic proline with the hydrophilic arginine (N13 in Table 1). In this case, the higher hydrophilicity of the mutant amino acid residue acts in the opposite direction: decrease (instead of increase) in the 3D structural stability because the substituted proline residue P491 is located in a crypt pocket of the protein globule (this is why the images of the substituted/substituting residues look colorless in the second row in Figure 7). Despite its deep location, the mutant amino acid residue has contact with water molecules; indications of this are the positive electric charge of the arginine residue and the shift in the isoelectric point with 0.2 pH units (N13 in Table 2).

The contribution of hydrophilicity to the decreased structural stability can be explained by the larger number of the surrounding water molecules and their orientation around the (positive) coulomb charge of the arginine amino group due to its strong electric field and the dipole moments of the H_2_O molecules. In cases when a deeply located chargeable group is isolated from the aqueous medium, it remains uncharged because of trapping or loss of a proton (COO^−^ → COOH or NH_3_^+^ → NH_2_) in the process of folding of the polypeptide chain upon its intracellular syntheses on the ribosome. These effects appear because of the much smaller dielectric permittivity (ε ≈ 2–4) of the core of the protein globule (where the hydrophobic amino acid residues are located) in comparison with that of the water medium (ε = 80). As a result, the electrostatic forces in the globule core are 20–40 times stronger than the forces on the globule surface.

The decrease in the 3D structural stability of the S1-subunit also occurs in the case of the mutant P491D when the hydrophobic uncharged proline residue is substituted with the negatively charged hydrophilic aspartate residue (N14 in Table 1 and Table 2). In this case, the decrease in the folding free energy is slightly less because the negative coulomb charge of the substituting residue emerges in a region with positive electrostatic potential (Figure 7, picture 491Asn). The influence of the electric effect is smaller, considering the small difference in ΔΔ*G*_fold_ of the two mutants with positive or negative charge of the substituting amino acid residues (N13 and N14 in Table 3) because of the deep location of the charged amino acid residues (the pictures 491Arg and 491Asn in Figure 7), which increases the distance between their coulomb charges and those located on the surface of the protein globule. However, the decrease in the folding free energy is relatively high (ΔΔ*G*_fold_ ≈ +11 kJ/mol, N14 in Table 3) owing to the contribution of the reverse hydrophilic effect, which is caused by the substitution of the hydrophobic amino acid residue with the hydrophilic one, both located in a hydrophobic environment under the surface of the protein globule.

#### 2.5.6. Deep Hydrophilic Effect

The reverse hydrophilic effect appears in clear form (without the electrostatic contribution) in mutant P491N (N15 in Table 1 and Table 2, the picture 491Asp in Figure 7) where the hydrophobic proline residue is substituted with the hydrophilic uncharged asparagine. In this case, the 3D structural stability decreases owing to the substitution of the hydrophobic residue with hydrophilic one located in a crypt pocket where the surrounding residues are hydrophobic. However, the alteration of the folding free energy in this case is relatively small (ΔΔ*G*_fold_ ≈ +4 kJ/mol). It could be supposed that the reason for this is that the reverse hydrophilic effect is compensated by the stabilization of the 3D structure of the S1-subunit. This possibility is based on the lower configurational entropy of the pyrrole ring, which stabilizes the protein structure [38]. The specific structure of the proline residues strongly limits the configuration angles, allowing the formation of an α-helix. As a result, the proline P491 ends the α-helix fragment (depicted as a ribbon in Figure 7, the upper row). The replacement of the proline residue with an asparagine one allows prolongation of the α-helix segment, but the reconstructed 3D structures do not confirm this expectation (the length of the ribbon remains unaltered).

### 2.6. Peculiar Properties of S1-Mutants

The pI value of the RBD of the mutant N354D (N2 in Table 1, Table 2 and Table 3), detected in China [39], is lower with 0.34 pH units compared to that of the wild-type strain of SARS-CoV-2. The negative ΔpI shift is caused by the replacement of the neutral asparagine (N) amino acid residue at position 354 in the polypeptide chain (numbered from the N-end to C-end) with the negatively charged aspartate (D) whose carboxylic group is fully dissociated at pH 6–7. This negative pI-shift means that the return to equality of the positive and negative charges appears at a 2.2-times-higher H^+^ concentration in the medium. The appearance of one negative charge on the surface with positive local potential leads to some stabilization of the 3D structure of the S1-subunit: the absolute value of the folding free energy increases with 1.5–2.2 kJ/mol at pH 6 and pH 7, i.e., in the extracellular secretion and the cytoplasm of the epithelial cells of the respiratory tract. The decrease in the positive surface electrostatic potential at the receptor-binding interface of the S1-subunit (the pictures 354Asp in Figure 3) decreases the S1-ACE2 association constant and, consequently, the infectivity of this coronavirus point mutant.

Just the opposite alterations appear in the mutant D364Y (N6 in Table 1, Table 2 and Table 3), isolated in China [40]: positive pI-shift with 0.23 pH-units for RBD and a decrease in the 3D structural stability of the S1-subunit (the free folding energy increases by 5–6 kJ/mol). These changes are caused by the substitution of the negatively charged hydrophilic aspartate (D) residue with the uncharged (at neutral pH) hydrophobic tyrosine (Y) at position 364. The disappearance of the negative coulomb charge leads to a change in the surface electrostatic potential from slightly negative to slightly positive (the pictures Asp364 and 364Tyr in Figure 4) at the receptor-binding interface and accordingly to a stronger association of the S-protein with the negatively charged ACE2 receptor (Figure 2) of the epithelial cells.

The picture 408Ile in Figure 5 shows the surface electrostatic potential at the receptor-binding interface of the mutant R408I (N9, in Table 1, Table 2 and Table 3), isolated in India [41], which becomes almost neutral upon substitution of the positively charged arginine (R) amino acid residue (whose guanidine group bears a bound H^+^ at neutral pH) with the uncharged residue of the isoleucine (I) amino acid. This predicts a weakened association of the S-protein with the ACE2 receptor (whose bounding interface is negatively charged, Figure 2). The negative shift ΔpI 0.36 in the isoelectric point is almost equal to that of the Chinese N354D mutant (N2 in Table 2), although the change is quite different: in the Indian mutant, one positive coulomb charge disappears, instead of one negative charge arising in N354D. However, the two mutations cause opposite alterations in the 3D stability of their S1-subunits: the hydrophobicity of the isoleucine residue leads to a decrease in the folding free energy (by its absolute value |Δ*G*_fold_|: ΔΔ*G*_fold_ = +9.2 kJ/mol for R408I instead of ΔΔ*G*_fold_ = −2.1 kJ/mol in the case of N354D) at pH 7. The value of |Δ*G*_fold_| includes both hydration and electrostatic components of the Gibbs free energy; the latter increases in both cases due to a reduction in the electrostatic repulsion in the region of the point mutation; an indication for this is the decrease in the (positive) local electric potential caused by the substitution of the uncharged asparagine residue with negatively charged aspartate, or the disappearance of one positive charge upon substitution of the arginine residue with the isoleucine one, respectively, at mutations N354D and R408I (the pictures 354Asp in Figure 3 and 408Ile in Figure 5); i.e., the S-protein of the R408I mutant is less stable in comparison with the wild-type strain and with the N354D mutant mainly because of a decrease in the dehydration energy upon replacement of one hydrophilic with a hydrophobic amino acid residue located on the surface of the S1-subunit.

In the case of the W436R mutant (N10, in Table 1, Table 2 and Table 3), isolated in China [42], the shift ΔpI = +0.26 of the isoelectric point is caused by replacement of the uncharged tryptophan (W) residue with the positively charged arginine (R) residue. The shift to a higher pI value is analogous to the disappearance of one negative charge in the D364Y mutant (N6) considered above. The addition of the positive coulomb charge on the surface of the S1-subunit in the region of the S1-ACE2 receptor-binding interface leads to a stronger virus–cell association because of the increased electrostatic potential (the picture 436Arg in Figure 6). This charge change is accompanied by a small decrease in the structural stability of the S1-subunit (ΔΔ*G*_fold_ = +2.2 kJ/mol) because of the stronger electrostatic repulsion, although the substitution of the hydrophobic tryptophan residue with the hydrophilic arginine one increases the hydration energy on the surface of the protein globule; i.e., in the case of the mutant W436R, the electrostatic component of the folding free energy Δ*G*_fold_ is somewhat greater than the hydrational one.

The shift in the isoelectric point (ΔpI = +0.25) of the RBD and the relatively significant reduction in the folding free energy of the S1-subunit (ΔΔ*G*_fold_ = 13–14 kJ/mol at pH 6–7) of the P491R mutant (N13, in Table 1, Table 2 and Table 3), isolated in USA [43], are caused by the substitution of the uncharged hydrophobic proline (P) amino acid residue of the wild-type strain with the positively charged hydrophilic arginine (R) residue. In this case, the electrostatic and hydrophilic components of the folding free energy Δ*G*_fold_ accumulatively decrease the 3D structural stability of the S1-subunit because of the location of the mutant amino acid residue under the surface of the protein globule, where it is surrounded by hydrophobic residues (reverse hydrophilic effect). The increased positive electrostatic potential on the receptor-binding surface of the S1-subunit (picture 491Arg in Figure 7) predicts a stronger association of the S-protein to the negatively charged ACE2 receptor, and, accordingly, a higher infectivity of this coronavirus mutant.

## 3. Discussion

### 3.1. Gibbs Free Energy upon Folding

The reduction in the Gibbs free energy Δ*G*_fold_ = *G*_N_ − *G*_R_ (where indexes N and R denote the 3D and the random coil conformations) upon folding or unfolding (Δ*G*_unfold_ = −Δ*G*_fold_) of a polypeptide chain has four components which determine the stability of the 3D structure of the protein globule: (a) the energy of hydration (the so-called ‘hydrophobic’ forces) which gives the main contribution (about 9/10) to the folding of the chain into a globule due to the inability of hydrophobic residues to form H-bonds with the surrounding water molecules; (b) the energy of intramolecular hydrogen bonds which fixes the secondary structure (α-helix and β-sheet) via H-bonds (C=O…H–N) between the electronegative O and N atoms of the polypeptide backbone; (c) the energy of the electrostatic attraction and repulsion between coulomb charges of the ionized groups (–NH_3_^+^ and –COO^–^) of the chargeable amino acid residues; and (d) van-der-Waals (London dispersive forces, dipole–dipole, and charge-induced dipole) interactions. The substitution of an amino acid residue with another (as a result of point mutation in the RNA chain of the coronavirus) can cause alterations in all four components of the Gibbs free energy Δ*G*_fold_ upon folding of the polypeptide chain.

The electric component of Δ*G*_fold_ alters its value upon substitution of a charged with an oppositely charged or uncharged amino acid residue, or vice versa. The quantity Δ*G*_fold_ depends on the pH of the medium because of the ionization of chargeable groups of nine types of amino acid residues. The main contributions come from the carboxylic groups of asparagine (pK_a_ 3.0–4.7) and glutamine (~pK 4.4) amino acids, the imidazole group of histidine (pK_a_ 5.6–7.0), the amino group of the lysine (pK_a_ 9.4–10.6), and the guanidine group of arginine (pK_a_ 11.6–12.6). The acid dissociation constant K_a_ (and its negative logarithm pK_a_) of a given group has an individual value [37], which is determined by its affinity for H^+^-ions and the local concentration of the hydroxonium cations H_3_O^+^; the latter is determined by the bulk pH and the local electrostatic potential which increases or decreases [H_3_O^+^], according to its negative or positive sign, respectively. The local electric field is a superposition of the elementary electric fields originating mainly from the neighboring coulomb electric charges. The contribution of the electric fields of the permanent and induced dipoles of neighboring uncharged groups and those of the distant coulomb charges is small because of the strong dependence on the distance. The Δ*G*_fold_ has similar values at pH 6 and pH 7 (Table 3) because the constant of dissociation pK_a_ of the negatively charged carboxylic groups and positively charged amino and guanidine (arginine) groups appears in the acidic and basic pH ranges, respectively. At the physiological pH of 6–7, only the pK_a_ of the imidazole (histidine) groups emerges. As a result, the degree of ionizations of the COO^–^ and NH_3_^+^ groups does not significantly alter within the physiologically important pH 6–7 range.

The hydrophobic component of the folding free energy Δ*G*_fold_ changes with every substitution of one amino acid residue with another due to their different affinity for the water molecules. A measure for the hydration energy is the alteration Δ*G*_trans_ of the Gibbs free energy upon translation of a given amino acid residue from an aqueous (polar) to a non-polar or less polar medium. The values of Δ*G*_trans_ upon the transition from ethanol (relative dielectric permittivity ε = 25 at 20 °C) to water (ε = 80) are given in Table 1. A residue is considered hydrophilic when Δ*G*_trans_ has a negative (−) sign, and hydrophobic when it has a positive (+) sign. The hydrophilicity/hydrophobicity is determined by the ratio of the total surfaces of the polar groups (containing O, N, S atoms) and the non-polar aliphatic (CH_2_) groups. When the mutant amino acid residue is located on the surface of the protein globule, the substitution of a hydrophilic residue with a hydrophobic one leads to a decrease in the absolute value |Δ*G*_fold_| of the folding free energy (positive increment +ΔΔ*G*_fold_, as defined by Equation (2)). Conversely, when the mutation emerges in the globule’s core (or in a crypt pocket), where the surrounding amino acid residues are hydrophobic, it leads to the reverse hydrophilic effect (Section 2.5.6).

### 3.2. Computing of the Folding Free Energy

The computer technique of protein folding [44,45] is based on the assumption that in the initial state, the polypeptide chain is in a random coil conformation in an aqueous medium, without any intra- and intermolecular interactions. This state is equivalent to that of an uncharged single polymer chain in a good solvent at approximately zero polymer concentration. The polypeptide chain spontaneously forms a protein globule with a 3D structure (due to a decrease in the Gibbs free energy). The computation of the folding free energy Δ*G*_fold_ takes into account the intramolecular interactions by summing up the energy of H-bonds, and hydrophobic and electrostatic forces, using the known energies of the single bonds and the accessibility to the water molecules of the medium. In the computation, the atom coordinates of the actual 3D structure are used, which are obtained experimentally using X-rays of the crystallized protein, NMR in aqueous solution, or cryo-EM or are calculated by in silico mutagenesis under the condition of minimal free energy.

We have estimated the accuracy of this technique by comparing the computed folding free energy Δ*G*_fold_ with the unfolding energy Δ*G*_unfold_ (the reverse quantity: Δ*G*_unfold_ = −Δ*G*_fold_) obtained experimentally upon denaturation from the native globule to the random coil in the water solution. For this purpose, we used the literature data for cytochrome *c* (a mitochondrial hemoproteid) which is comprehensively investigated. We calculated Δ*G*_fold_ under conditions corresponding to the experiment: Δ*G*_fold_ = −61 kJ/mol in 150 mM NaCl (physiological solution), Δ*G*_fold_ = −59 kJ/mol in 50 mM NaCl (used in the calorimetric experiments), and Δ*G*_fold_ = −43 kJ/mol in 0.1 mM NaCl. These values of Δ*G*_fold_ are in satisfactory agreement with the unfolding free energy Δ*G*_unfold_ = +(40–70) kJ/mol measured by different experimental methods: differential calorimetry [46,47], optical [48], and combination of both techniques [49]. Therefore, the quite satisfactory coincidence of the absolute values |−Δ*G*_fold_| and |Δ*G*_unfold_| obtained for cytochrome *c* gives us confidence to apply the computer programs for estimation of the 3D structural stability of other globular proteins by calculation of the Gibbs free energy upon folding of their polypeptide chains. 

### 3.3. Protein–Receptor Attraction

When the pH of the medium is below the pI, protein macromolecules are predominantly positively charged. At pH 6–7, the net charge of the S1-subunit of the considered mutants is positive because their isoelectric point is within the range pI 8.2–9.0 (Table 2). This finding can be expended to all possible point mutants, except for coronavirus variants with multiple substitutions of charged amino acid residues, in particular those emerging upon deletion of a part of the polypeptide chain. This means that the positively charged S-protein will have a more or less strong electrostatic attraction to the negatively charged ACE2 receptor of the epithelial cells, leading to the association of the two proteins. This inference is well founded in the absence of salts in the medium, where the low ionic strength determines the strong electrostatic interaction between the two protein globules. However, at the physiological concentration of NaCl (0.15 mol/L) in the blood plasma, the electrostatic S-ACE2 attraction predominantly appears in the relatively small area of the S-ACE2 interface owing to the shielding of protein charges by the counterions in the medium. Consequently, the surface electrostatic potential is short-distance-acting. That is why we have considered only the point mutations which emerge on the surface of the S1-subunit at the receptor-binding interface. The computer reconstruction of the 3D structure of the RBD of the S-protein, using the published amino acid sequence of new point mutants or coronavirus variants, enables the computation of the local surface electrostatic potential and prediction of their infectivity.

The results (Section 2.4) have revealed that the local electrostatic potential remains more or less positive in most cases (the bottom rows in Figure 3, Figure 4, Figure 6 and Figure 7). Therefore, these point mutants could cause coronavirus infection, especially in the upper respiratory tract where the ionic strength is much lower than in the blood plasma. The most dangerous could be those point mutations where a negatively charged amino acid residue is substituted with a positive one, as in the case of mutant D364R (N4 in Table 2, the picture 364Arg in Figure 4). In contrast, the point mutant R408D (N7, the picture 408Asp in Figure 5), where a positive charge is substituted with a negative one, could not cause infection under physiological conditions, because the local electrostatic potential has a negative sign at the receptor-binding interface.

## 4. Methods

The following programs were used: (a) Site-Directed Mutator (SDM) for in silico mutagenesis to find the atom coordinates of a mutant model with minimal free energy by analogy with 3D local structures of other native proteins [50,51]; (b) PHEMTO [52] and Propka [53,54] for protein electrostatics to calculate the surface electric potential; (c) Bluues [55] for protein folding energy; (d) SuperPose for macromolecular alignment for comparison of 3D structure of the original and mutant proteins; and (e) Chimera [56], PBEQ Solver [57], and VMD: Visual molecular dynamics 1.9.2 [58] for visualization of molecular models and the electrostatic potential on the protein globule surface.

The surface electric potential of the ACE2 receptor was computed using its crystallographic structure. The analysis of the S1-subunit of the virus mutants was made in five steps: (a) selecting point mutants in whose polypeptide chain an uncharged amino acid residue is replaced with a positively or negatively charged residue with the same or opposite hydrophilicity/hydrophobicity, or vice versa; (b) computer reconstruction of the 3D structure of the S1-subunit of the selected mutants on the base of the crystallographic structure of the S1-subunit of the wild-type coronavirus; (c) selecting 3D models in which a replaced amino acid residue with different charge or hydrophilicity is situated on the surface of the RBD at the interface of the association of the S1-subunit with the ACE2 receptor; (d) calculating the electrostatic parameters of the S1-subunit of the mutants: pK values of the ionizable groups, pH dependence of the total charge, the isoelectric point (zero net charge), the electrostatic component of the free energy; (e) calculating the 3D electrostatic potential of the reconstructed S1-subunits at a given pH, and (f) visualizing the 2D potential on the surface of the protein globule.

## 5. Conclusions

The single amino acid substitution at a point mutation in the S1-subunit of the S-protein leads to alterations in its 3D structural stability, which are caused by both the electric charge and hydrophilicity of the replaced and mutant amino acid residues. However, these alterations are relatively small, and the S-protein remains stable at the physiological pH of 6–7; an indication for this is the negative sign of the folding free energy Δ*G*_fold_. The addition or disappearance of even one coulomb electric charge causes a noticeable shift in the isoelectric point pI up to one pH unit for the RBD of the S1-subunit. At pH 6–7, the S1-subunit is positively charged because its isoelectric point lies in the pI range of 8–9 in all cases of single point mutations. This determines its electrostatic association with the negatively charged ACE2 receptor of the epithelial cells. Under physiological conditions of high salt concentrations (high ionic strength), the electrostatic attraction acts within a short distance because of counterion shielding. Therefore, the S-ACE2 association is determined by the local electrostatic potential at the interface of the two protein globules. At some point mutations, the value of the local potential on the surface of the S1-subunit at the receptor-binding interface undergoes dramatic alterations in the vicinity of the mutant amino acid residue. This predicts a decrease or an increase in the electrostatic component of the S-ACE2 association energy, respectively, upon substitution of a positively charged with a negatively charged or uncharged amino acid residue, or vice versa. This allows prediction of the relative infectivity and contagiousness (under equal, other conditions determining the social immune status) of new SARS-CoV-2 mutants with the determined amino acid sequence by reconstruction of their 3D structure and calculation of the surface electrostatic potential whose local value can be used as a criterion appropriate for predicting alterations in the infectivity, instead of relying on the weakly sensitive isoelectric point of the S-protein, S1-subunit, or RBD.

## Figures and Tables

**Figure 1 ijms-25-02174-f001:**
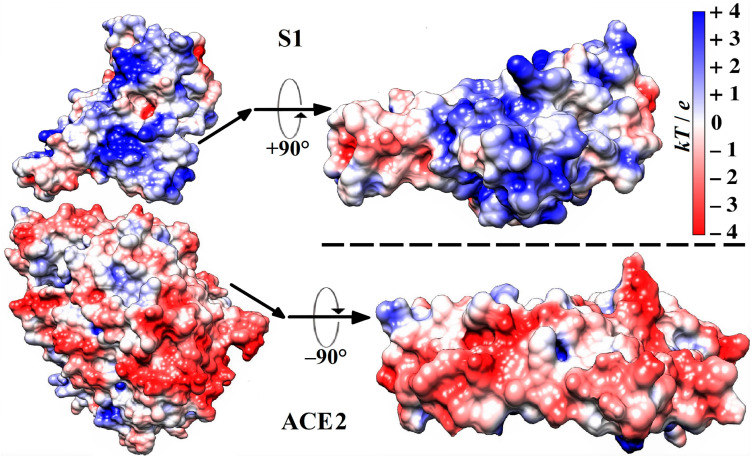
Molecule models (all C, O, N, and H atoms are included) of a section of the S1-ACE2 complex (according to its crystallographic structure deposited in Protein Data Bank: 6LZG) in the region of the receptor-binding domain (RBD) (on the **left**), and the contact surfaces (on the **right**) by which the human ACE2 receptor associates with the S1-subunit of the wild-type strain of SARS-CoV-2 β-coronavirus. The two protein globules on the right are rotated by −90° (S1) and +90° (ACE2) to display the receptor-binding and the virus-binding contacting surfaces, respectively. The electrostatic potential φ = *kT*/*e* on the surfaces of the two globular proteins is computed using their atomic coordinates and visualized by coloration according to its sign and value in blue (positive), red (negative), and white (neutral) with scale *kT*/*e* = ±4 J/C (*k*—Boltzmann constant [J/K], *T*—absolute temperature [K], *e*—charge of the proton [C]; 1 *kT*/*e* [J/C] = 26.7 mV at 37 °C).

**Figure 2 ijms-25-02174-f002:**
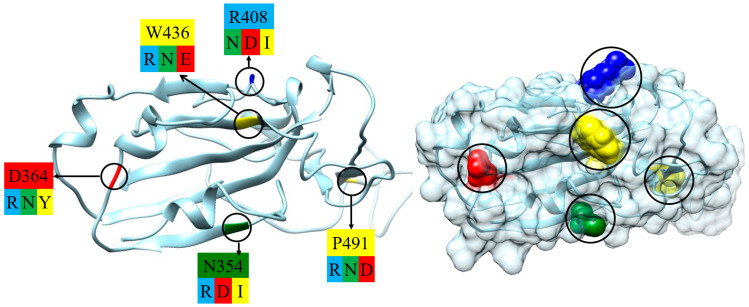
Part of the S1-subunit of the wild-type strain of coronavirus (segment which forms receptor-binding domain, RBD): skeletal model (polypeptide backbone, (**left**) picture) and its surface ((**right**) picture) according to the crystallographic structure [PDB: 6LZG]; the point mutations are denoted by cycles. The substituted amino acid residues are colored according to their electric charge and hydrophilicity/hydrophobicity: blue (positively charged, hydrophilic), red (negatively charged, hydrophilic), green (uncharged, hydrophilic), and yellow (uncharged, hydrophobic). The charge/hydrophilicity of the substituting (mutant) amino acid residues is indicated by the same colors under the type/number of the substituted residues ((**left**) picture); the type of the amino acid residues is denoted by the one-letter code (Table 1).

**Figure 3 ijms-25-02174-f003:**
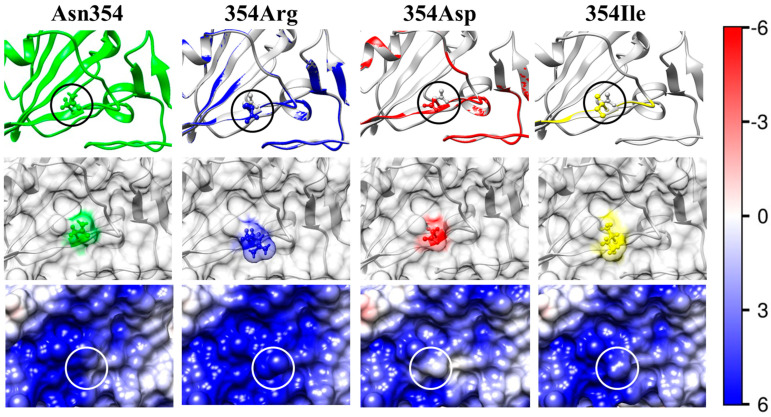
Structural models and surface electrostatic potential of the wild-type strain (first column, Asn354) and the mutants (second, third, and fourth columns) of the RBD of the S-protein. The pictures show a part of the 3D structure and the surface of RBD in the region of the point mutations (denoted by rings in the first and third rows). The uncharged hydrophilic amino acid residue of the asparagine (Asn, N) at 354th position (numbered from the N-end to the C-end of the polypeptide chain) (first column) of the wild-type S-protein is substituted with the following: (354Arg) a positively charged hydrophilic residue, arginine (Arg, R; the second column); (354Asp) a negatively charged hydrophilic residue, aspartic acid (aspartate, Asp, D; the third column); or (354Ile) an uncharged hydrophobic residue, isoleucine (Ile, I) (the fourth column). The amino acid residues in the second row are colored according to their charge and hydrophilicity: green (uncharged hydrophilic), blue (positively charged hydrophilic), red (negatively charged hydrophilic), and yellow (uncharged hydrophobic). The first (upper) row of pictures shows skeletal models of segments of the polypeptide chain backbone; the unstructured and α-helix segments are depicted, respectively, as curves and arrow-ribands directed from the N to C end of the polypeptide chain. The first picture in the upper row shows a fragment of the wild-type RBD whose amino acid residue (in the ring) has undergone mutations shown in the right three pictures. The second, third, and fourth pictures in the upper row show the 3D aliment of the skeletal models of the wild-type strain and the point mutants. Discrepant segments (whose 3D coordinates are shifted) of the mutant polypeptide chain are colored according to the charge and hydrophilicity of the substituting (mutant) amino acid residue against the background of the gray polypeptide chain of the wild-type strain. The four pictures in the lower row represent the electric potential on the surface of RBD at pH 7.0. The potential is visualized by colorings according to its sign: positive (blue), negative (red), and neutral (white); the intensity of the colors corresponds to the *kT/e* scale (shown on the right) in the range ± 6 *kT*/*e* [J/C]; 1 *kT*/*e* = 26.7 mV at 37 °C.

**Figure 4 ijms-25-02174-f004:**
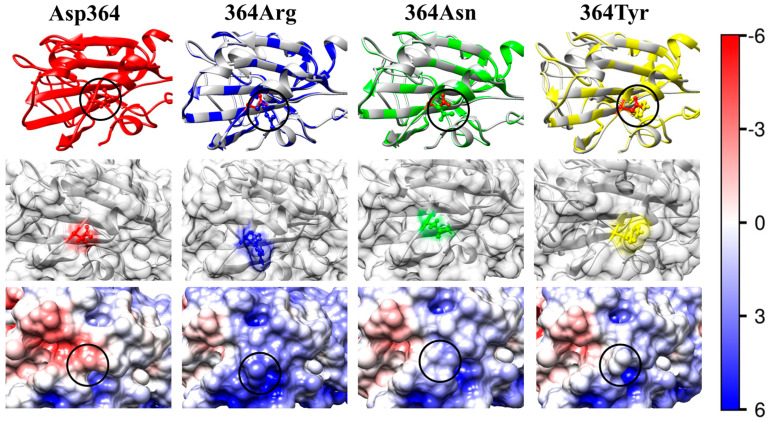
The same as in Figure 3 but in cases when the negatively charged hydrophilic amino acid residue of aspartic acid (aspartate, Asp, D) at 364th position in the wild-type RBD (the left column Asp364) is substituted with mutant residues of arginine (Arg, R; positively charged hydrophilic; the second column 364Arg), asparagine (Asn, N; uncharged hydrophilic, the third column 364Asn), or tyrosine (Tyr, Y; uncharged hydrophobic; the fourth column 364Tyr).

**Figure 5 ijms-25-02174-f005:**
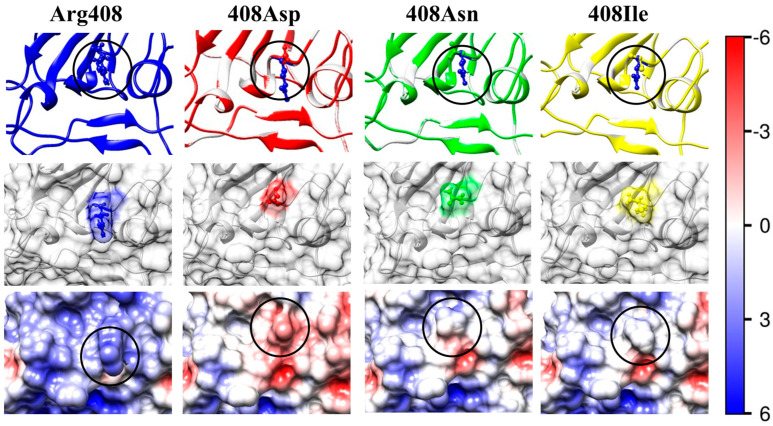
The same as in Figure 3 but the positively charged hydrophilic arginine (Arg, R) residue at 408th position in the wild-type RBD (the first column Arg408) is substituted with mutant residues of aspartic acid (aspartate, Asp, D; negatively charged hydrophilic; the second column 408Asp), asparagine (Asn, N; uncharged hydrophilic, the third column 408Asn), or isoleucine (Ile, I; uncharged hydrophobic; the fourth column 408Ile).

**Figure 6 ijms-25-02174-f006:**
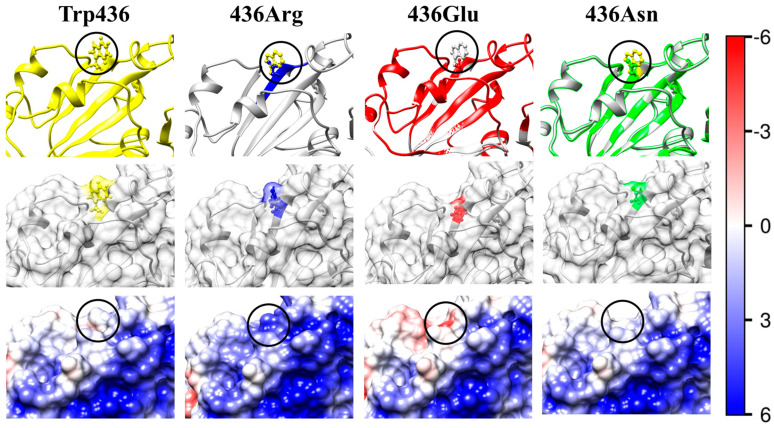
The same as in Figure 3 but the uncharged hydrophobic tryptophan (Trp, W) residue at 436th position in the wild-type RBD (the left column Trp436) is substituted with mutant residues of arginine (Arg, R; positively charged hydrophilic; the second column 436Arg), glutamine (glutamine acid, Glu, E, negatively charged hydrophilic, the third column), or asparagine (Asn, N; uncharged hydrophilic; the fourth column 436Asn).

**Figure 7 ijms-25-02174-f007:**
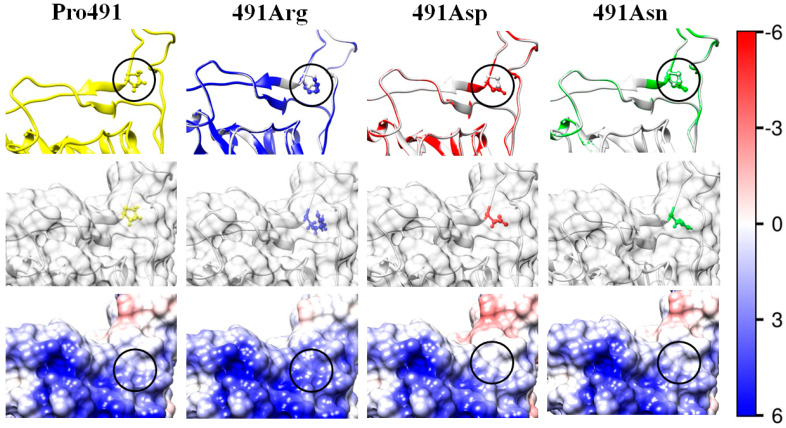
The same as in Figure 3 but the uncharged hydrophobic proline (Pro, P) residue at 491th position in the wild-type RBD (the left column Pro491) is substituted with mutant residues of arginine (Arg, R; positively charged hydrophilic; the second column 491Arg), aspartic acid (aspartate, Asp, D, negatively charged hydrophilic, the third column 491Asp), or asparagine (Asn, N; uncharged hydrophilic; the fourth column 491Asn). The substituted (initial) and the substituting (mutant) amino acid residues are in crypt pocket under the surface of the protein globule; as a result, the residues are colorless (the second row).

**Figure 8 ijms-25-02174-f008:**
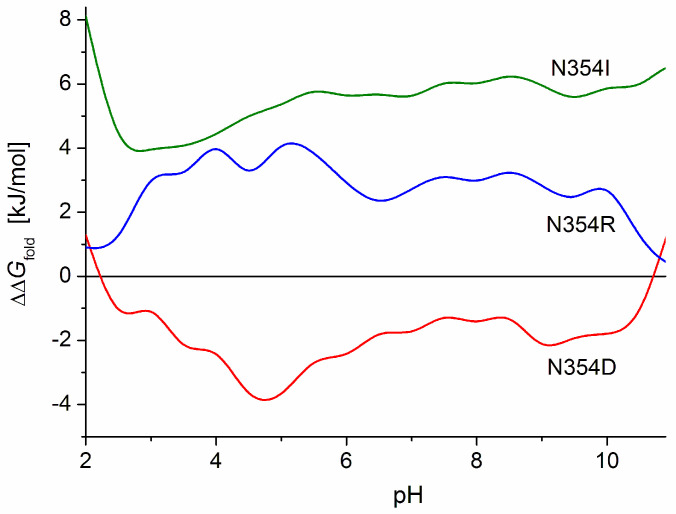
pH dependence of the difference ΔΔ*G*_fold_ in Gibbs free energy of folding Δ*G*_fold_ of the polypeptide chain of the S1-subunit of the S-protein in point mutant N354 and the wild-type strain upon substitution of the asparagine (N, hydrophilic uncharged) amino acid residue at position 354 with aspartate (D, negatively charged hydrophilic, curve N354D), arginine (R, positively charged hydrophilic, curve N354R), or isoleucine (I, hydrophobic uncharged, curve N354I).

**Table 1 ijms-25-02174-t001:** Mutant amino acid residues with molecular mass *M*_res_ [g/mol] [35], volume *V*_res_ [nm^3^], and alteration Δ*G*_trans_ [kJ/mol] of Gibbs free energy upon transfer from ethanol to water [34] at pH 7, and increment ΔΔ*G*_trans_ [kJ/mol] in the free energy upon substitution of a given amino acid residue with another (both localized on the surface of the S1-subunit of the S-protein of SARS-CoV-2 coronavirus).

Mutant		Replaced Residue				Substituting Residue				ΔΔ*G*_trans_
№	Code	Name	*M* _res_	*V* _res_	Δ*G*_trans_	Name	*M* _res_	*V* _res_	Δ*G*_trans_	
1	N354R	asparagine	114	0.15	−3.3	arginine	157	0.21	−6.3	−3.0
2	N354D	asparagine	114	0.15	−3.3	aspartate	114	0.15	−4.6	−1.3
3	N354I	asparagine	114	0.15	−3.3	isoleucine	113	0.21	+10.0	+13.3
4	D364R	aspartate	114	0.15	−4.6	arginine	157	0.21	−6.3	−1.7
5	D364N	aspartate	114	0.15	−4.6	asparagine	114	0.15	−3.3	+1.3
6	D364Y	aspartate	114	0.15	−4.6	tyrosine	163	0.15	+5.4	+10.0
7	R408D	arginine	157	0.21	−6.3	aspartate	114	0.15	−4.6	+1.7
8	R408N	arginine	157	0.21	−6.3	asparagine	114	0.15	−3.3	+3.0
9	R408I	arginine	157	0.21	−6.3	isoleucine	113	0.21	+10.0	+16.3
10	W436R	tryptophan	186	0.25	+12.6	arginine	157	0.15	−6.3	−18.9
11	W436E	tryptophan	186	0.25	+12.6	glutamate	128	165	−1.3	−13.9
12	W436N	tryptophan	186	0.25	+12.6	asparagine	97	0.13	−3.3	−8.4
13	P491R	proline	97	0.13	+4.2	arginine	157	0.21	−6.3	−10.5
14	P491D	proline	97	0.13	+4.2	aspartate	114	0.15	−4.6	−8.8
15	P491N	proline	97	0.13	+4.2	asparagine	114	0.15	−3.3	−7.5

**Table 2 ijms-25-02174-t002:** Isoelectric point (IEP) pI of the S1-subunit and RBD of the S-protein with one amino acid residue substituted, charged positively (P), negatively (N), or neutral (0) at pH 7, and difference ΔpI in IEP of the mutant and the wild-type strain of SARS-CoV-2 coronavirus.

Mutant		Replaced Residue		Substituting		S1-Subunit		RBD	
№	Code	Name	+/−	Name	+/−	pI	ΔpI	pI	ΔpI
0	Wild	–	0	–	0	8.70	0	9.02	0
1	N354R	asparagine	0	arginine	P	8.87	+0.17	9.27	+0.25
2	N354D	asparagine	0	aspartate	N	8.50	−0.20	8.68	−0.34
3	N354I	asparagine	0	isoleucine	0	8.70	0	9.01	−0.01
4	D364R	aspartate	N	arginine	P	9.01	+0.31	9.48	+0.46
5	D364N	aspartate	N	asparagine	0	8.87	+0.17	9.28	+0.26
6	D364Y	aspartate	N	tyrosine	0	8.86	+0.17	9.25	+0.23
7	R408D	arginine	P	aspartate	N	8.21	−0.49	8.07	−0.95
8	R408N	arginine	P	asparagine	0	8.49	−0.21	8.66	−0.36
9	R408I	arginine	P	isoleucine	0	8.49	−0.21	8.66	−0.36
10	W436R	tryptophan	0	arginine	P	8.87	+0.17	9.28	+0.26
11	W436E	tryptophan	0	glutamate	N	8.49	-0.21	8.65	−0.37
12	W436N	tryptophan	0	asparagine	0	8.70	0	9.02	0
13	P491R	proline	0	arginine	P	8.87	+0.17	9.27	+0.25
14	P491D	proline	0	aspartate	N	8.49	−0.21	8.66	−0.36
15	P491N	proline	0	asparagine	0	8.7	0	9.02	0

**Table 3 ijms-25-02174-t003:** Gibbs free energy of folding Δ*G*_fold_ of S1-subunit of S-protein and the difference ΔΔ*G*_fold_ of Δ*G*_fold_ values of mutant and wild-type strain of SARS-CoV-2 coronavirus.

S1-Subunit		pH 6		pH 7		pH 9	
№	Mutant	Δ*G*_fold_ [kJ/mol]	ΔΔ*G*_fold_ [kJ/mol]	Δ*G*_fold_ [kJ/mol]	ΔΔ*G*_fold_ [kJ/mol]	Δ*G*_fold_ [kJ/mol]	ΔΔ*G*_fold_ [kJ/mol]
0	Wild	−95.17	0	−122.47	0	−138.28	0
1	N354R	−92.30	2.87	−119.47	2.73	−135.46	2.82
2	N354D	−96.69	−1.52	−124.62	−2.15	−140.97	−2.69
3	N354I	−89.60	5.57	−117.00	5.47	−132.29	5.99
4	D364R	−85.79	9.38	−113.54	8.93	−129.21	9.07
5	D364N	−88.94	6.23	−117.21	5.26	−132.91	5.37
6	D364Y	−88.67	6.5	−117.39	5.08	−133.39	4.89
7	R408D	−85.08	10.09	−113.49	8.98	−129.86	8.42
8	R408N	−84.26	10.91	−111.97	10.5	−128.97	9.37
9	R408I	−84.03	11.14	−113.28	9.19	−129.43	8.85
10	W436R	−91	4.17	−120.26	2.21	−136.59	1.69
11	W436E	−85.24	9.93	−113.37	9.91	−128.98	9.3
12	W436N	−90.25	4.92	−118.87	5.75	−133.87	7.1
13	P491R	−80.88	14.29	−109.64	12.83	−123.3	14.98
14	P491D	−83.64	11.53	−111.85	10.62	−128.44	9.84
15	P491N	−90.83	4.34	−119.01	3.46	−134.33	3.95

## Data Availability

Data contained within the article.

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
