# Peer review of "Three-Dimensional Structural Stability and Local Electrostatic Potential at Point Mutations in Spike Protein of SARS-CoV-2 Coronavirus"

_ijms, 2024, doi:10.3390/ijms25042174_

Round 1
Reviewer 1 Report
Comments and Suggestions for Authors
The authors reconstructed the 3D structure of the S1 subunit of a selected coronavirus variant based on the published crystallographic structure of the S protein of the wild-type coronavirus and the amino acid sequence of the polypeptide chain of the mutant S proteins and then, after reconstructing the 3D structure of the mutants, calculated the isoelectric point, the electrostatic surface potential and the free folding energy of the S subunit at a certain pH of the medium.
Although there are many reports of calculations with RBD mutants, I have not found a series like this. The work is decent and the results and conclusions sound clear.
Comments on the Quality of English LanguageThe English language needs some work - e.g.:
- lines 40-42: this is confusing. Please make it 2 short sentences,
- line 124: "form" - should be "from",
- line132 - double "we"...
Author Response
We thank the Reviewer-1 for the good appreciation of our work and for the recommendations; they are introduced in the revised manuscript.
Reviewer 2 Report
Comments and Suggestions for Authors
In the manuscript presented, the authors substantiate an idea that electrostatic association determines the relative infectivity and contagiousness, which can be predicted by reconstruction of their 3D structure and calculation the surface electrostatic potential, instead of applying an isoelectric point for this prediction.
The paper is rather well written and logically substantiated. On the other hand, in the Introduction the authors incorrectly describe the mechanism of ACE2 operation: “… the most investigated is the angiotensin-converting enzyme (ACE2) whose main function is regulation of the blood pressure by detachment of two aminoacid residues from the peptide angiotensin-2 …”. But the truth is that ACE2 cleaves a single amino acid from either the angiotensin 1 decapeptide or the angiotensin 2 octapeptide to form a nona- and heptapeptide, respectively.
Also, English text needs to be edited: "instead" should be written "instead of" throughout the text; excessive use of semicolons in long sentences when one should use a period and formulate an idea using a series of short sentences; a lot of typos ("we we" on line 132, etc.)
Comments on the Quality of English Language"instead" should be written "instead of" throughout the text; excessive use of semicolons in long sentences when one should use a period and formulate an idea using a series of short sentences; a lot of typos ("we we" on line 132, etc.)
Author Response
We thank the Reviewer-2 for the good appreciation of our work and for the recommendations, especially about the function of the ACE2 (the first paragraph in Introduction). The recommended changes are made in the revised manuscript.